# Analysis of the spatio-temporal coupling coordination mechanism supporting economic resilience and high-quality economic development in the urban agglomeration in the middle reaches of the Yangtze River

Shenglan Ma, Junlin Huang[ID]*

School of Geographical Sciences, Hunan Normal University, Changsha, China

* huangjl1020@163.com

**Data Availability Statement:** The source of all basic data is presented in Table 2 of the

## Abstract

The coupling coordination mechanism between economic resilience and high-quality economic development is clarified not only to improve the capacity of economies to withstand external shocks but also to boost the level of sustainable economic development. However, economic resilience and economic quality research are still in their infancy. This study constructs an economic resilience indicator system based on the data of 28 prefecture-level cities in the urban agglomeration in the middle reaches of the Yangtze River (UAMRYR) from 2010–2019. The entropy approach was then coupled with the coupling coordination model to determine the level of coupling coordination between the two. Finally, the geographical detector was employed to reveal the coupling coordination process and driving elements. This study demonstrates that: (1) The economic resilience and high-quality economic development of the UAMRYR are on the rise; (2) The coupling coordination degree between economic resilience and high-quality economic development is improving, with the average value increasing from 0.33 in 2010 to 0.46 in 2019, exhibiting a Core-Periphery spatial distribution pattern with Wuhan, Changsha, and Nanchang as the core; (3) The economic resilience resistance subsystem plays a substantial role in the driving influence of the coupling coordination degree, where the total retail sales of consumer goods indicator have an explanatory power of 0.8 or higher for coupled coordination. Innovation and inclusiveness thoroughly drive the coupling coordination degree among the five subsystems of high-quality economic development, where the q-value of the number of urban basic endowment insurance participants index is at least 0.9. Additionally, the interplay of the components demonstrates a complementary enhancement impact and a two-factor enhancement effect; (4) Economic resilience and high-quality economic development are complementary and provide a powerful incentive for the sustainable development of urban economic systems to achieve higher quality, more efficient, more egalitarian, and more sustainable economic development. Overall, this study improves the analysis of mechanism between economic

manuscript, and we have provided the relevant URLs. The part can be found on line 216.

**Funding:** This research was funded by Young Scientists Fund (52008167). The funders had no role in study design, data collection and analysis, decision to publish, or preparation of the manuscript.

**Competing interests:** The authors have declared that no competing interests exist.

resilience and high-quality economic development, and it provides more targeted guidance for economic development in the UARMRYR.

## Introduction

The 19th Communist Party of China National Congress declared that China's economy has entered the stage of high-quality economic development, and the 2018 Central Economic Work Conference proposed specific requirements for achieving high-quality economic development, indicating China's direction toward high-quality economic development. Concurrently, the globe is becoming increasingly insecure and uncertain. However, the relationship between economic resilience and high-quality economic growth is in its infancy, and additional research is necessary to discover how economic resilience can contribute to high-quality economic development. Therefore, researching the relationship between "Resilience" and "High-Quality Development" is of considerable practical importance in the current complicated and ever-changing international and local context.

Since 1973, when Holling [1] introduced the notion of "resilience" from engineering mechanics to the study of ecosystem restoration, the concept's content and scope have been developing. Reggiani A and Graaff T D [2] introduced the notion of resilience to the study of spatial economics for the first time; since then, resilience has been utilized in the field of economics. Academics have not yet achieved a consensus over the meaning of economic resilience. From a research standpoint, economic resilience can be subdivided into engineering resilience, ecological resilience, and evolutionary resilience; from a research methodology standpoint, it can be separated into unidimensional indicators and multidimensional indicators. Some researchers use regional sensitive index to measure economic resilience [3, 4]. Others, like Martin [5], define economic resilience as resistance, recovery, re-orientation and renewal. Economic systems are characterized by disequilibrium, dynamic evolution, and complicated adaptation; hence, it is currently more acceptable to define economic resilience in various dimensions. This study focuses on economic resilience, which is the capacity of the economic system to strengthen its capacity to withstand external shocks. Therefore, economic resilience is defined as the resistance of the regional economic system to preserve its stability after a shock, the recovery to return to the original growth pattern, and the evolution to adapt and develop based on its resources. Economic systems are characterized by disequilibrium, dynamic evolution, and complicated adaptation; hence, it is currently more acceptable to define economic resilience in various dimensions. This study focuses on economic resilience, which highlights the capacity of an economic system to increase its resistance to external shocks. Consequently, economic resilience is described as the ability of a regional economic system to sustain its stability following a shock, to revert to its original growth trajectory, and to evolve and develop by relying on its endowments. Martin [5] proposed a comprehensive analysis framework of regional economic resilience influencing factors in the fields of industrial structure, labor force, finance, and institutions, and Lagravinese, Di and other researchers [6, 7] conducted empirical studies in terms of industrial specialization level and policy system.

High-quality development is also known as high-quality economic development, as it focuses on the economic area. High-quality economic development highlights its systemic and comprehensive nature, which necessitates not only more efficient economic development and a more rational industrial structure but also better standards for green and low-carbon development [8–10]. Three categories comprise the discussion on the connotation of high-quality

economic development. The first category is based on the New Concept for Development perspective and the primary social conflicts [11, 12]. The second category emphasizes the shift of the economy from high speed to high quality [13, 14]. It is based on high-quality development. The third category distinguishes between the differing requirements of narrow and broad or micro and macro [15]. Regarding the measurement of the level of high-quality economic development, some scholars utilize total factor productivity to evaluate the amount of high-quality economic development [16], but the majority of studies establish a comprehensive system of indicators [17, 18]. The New Concept for Development [19, 20] is the primary source for the analysis of the influencing elements, but several researchers also examine the impact of other factors on the quality of economic development from a variety of perspectives. Tang, Zhang, and other experts [21, 22] suggest that enhancing the quality of urbanization can significantly boost high-quality economic growth. Xiao, Kong, and colleagues [23] identify the influence of innovation-driven strategy on high-quality development.

Domestic and international research on the mechanisms of economic resilience and high-quality economic development must be expanded. According to Chen and Liang [24], economic resilience is the basis for China's economy to remain within a sustainable operating range and define the country's continuous growth. Wang and other researchers [25, 26] suggest that it is essential to consistently promote high-quality economic transformation and upgrading through bolstering economic resilience. Li [27] argues that establishing a robust and dynamic capital market can contribute to the promotion of high-quality development. Sun and Yuan [28] use the western area to demonstrate that economic resilience strongly contributes to high-quality economic development in the near term, whereas the two have a long-term causal link. Guan and Zhang [29] find that economic resilience can contribute to high-quality economic development in the Yangtze River Delta region.

Existing research on economic resilience and high-quality economic development has made some progress, but the following limitations remain: on the one hand, in terms of research perspectives, most existing studies are limited to the one-way correlation between high-quality development and economic resilience or economic resilience to high-quality development, but there are few studies on the coupling and coordination of the synergistic interaction and two-way feedbacks between the two; on the other hand, the research content focuses on the measurement indicators or influencing factors of economic resilience or high-quality development, but the issues of how to promote the synergistic and balanced development of both and the drivers of economic resilience and high-quality economic development have not been fully demonstrated. Urban agglomeration in the middle reaches of the Yangtze River (UAMRYR) is an economic growth pole for the realization of the Central Rise, and the development of UAMRYR will significantly support the development the China [30]. It is crucial to examine the coupling mechanism between its economic resilience and high-quality development to enhance the region's sustainable development [31]. Consequently, based on the data of 28 prefecture-level cities in the UAMRYR from 2010–2019, this study uses the entropy method, coupling coordination model, and geographic detector to analyze the coupling coordination relationship between the two, to provide a reference for the path selection and policy formulation of high-quality economic development in the UAMRYR, and to supplement the research on the economic development of urban agglomeration in the central region.

## Materials and methods

### Methods

**The entropy method.**    We began by standardizing the initial data. Then, weights are assigned using the entropy approach. Finally, we utilized the multi-objective weighted

summation approach to calculate the complete evaluation value of economic resilience and the level of high-quality economic development in 28 prefecture-level cities in the UAMRYR. The formula is as follows:

$$P_{hi} = \sum_{j=1}^{z} Y_{hij} \times W_j$$

where $P_{hi}$ is the composite score of economic resilience and economic quality development level, $Y_{hij}$ and $W_j$ are the indicators and their respective weights after standardization.

**Coupling coordination degree model.** Coupling as a physical notion is utilized in a range of study domains [32] because of its versatility. The phenomenon of coupling is the interaction and influence between two or more systems, and it is frequently defined by the coupling degree [33]. The coupling degree model can be upgraded to the coupling coordination model (CCDM) which is designed to indicate whether the elements have a healthy co-development relationship. The formulas are as follows:

$$C = \left[ \frac{Y_1 \times Y_2}{[(Y_1 + Y_2)/2]^2} \right]^{\frac{1}{2}}$$

$$D = \sqrt{C \times T} \quad T = \alpha Y_1 + \beta Y_2$$

where, $D$ represents the degree of coupling coordination, and $D \in [0,1]$; $C$ represents the degree of coupling between economic resilience and high-quality economic development; $T$ reflects the overall level of economic resilience and high-quality economic development; $\alpha$ and $\beta$ in the formula are generally taken as 0.5 [34, 35], and $Y_1$ and $Y_2$ are the comprehensive indices of economic resilience and high-quality economic development, respectively. This study used the "0.1 cut-off point method" to classify them into 10 levels drawn on the research results of Ma and other scholars [36, 37], and the results are shown in Table 1:

**Geographical detector.** The geographical detector (GD) is a statistical tool for identifying spatial variability in geographical objects and understanding their driving mechanisms [38]. This method is used to explore the driving factors that determine the coupling coordination degree between economic resilience and high-quality economic development. The equation is as follows:

$$q = 1 - \frac{\sum_{h=1}^{L} N_h \sigma_h^2}{N \sigma^2} = 1 - \frac{SSW}{SST} \quad (h = 1, 2, 3 \ldots L)$$

where $q$ falls between [0–1], the bigger the value, the stronger the explanatory ability of the driver for the coupling coordination degree between economic resilience and high-quality economic development. The stratification of each indicator factor is denoted by $h$. $N_h$ and $N$ are

**Table 1. Classification of coupling coordination levels.**

| Value of D | Degree | Value of D | Degree |
|---|---|---|---|
| 0.0000–0.1000 | Extreme incoordination | 0.5001–0.6000 | Reluctant coordination |
| 0.1001–0.2000 | Severe incoordination | 0.6001–0.7000 | Primary coordination |
| 0.2001–0.3000 | Intermediate incoordination | 0.7001–0.8000 | Intermediate coordination |
| 0.3001–0.4000 | Mild incoordination | 0.8001–0.9000 | Favorable coordination |
| 0.4001–0.5000 | Basic incoordination | 0.9001–1.0000 | Quality coordination |

the sample sizes for stratum h and the region as a whole, respectively; $\sigma_h^2$ and $\sigma^2$ are the variances for stratum h and the overall region, respectively.

## Model building

**Study area and data sources.**   The urban agglomeration development plan for the Middle Reaches of the Yangtze River points out that UAMRYR is an extremely large state-level urban agglomeration, located in the transition zone between China's second and third terrain tiers. UAMRYR is a regional center consisting of three main metropolitan regions, namely Wuhan urban agglomeration, Chang-Zhu-Tan urban agglomeration, and Poyang Lake urban agglomeration, and encompassing 28 prefecture-level cities in Hubei, Hunan, and Jiangxi provinces. Based on the planning scope of the Development Planning of the Urban Agglomeration in the middle reaches of the Yangtze River and taking data completeness into account, the scope of this study encompasses Fuzhou and Ji'an of Jiangxi Province as a whole, as well as 28 prefecture-level cities in Hubei, Hunan, and Jiangxi provinces. based on the planning scope of the Development Planning of the Urban Agglomeration in the Middle Reaches of the Yangtze River and considering the completeness of data. From 2010–2019, panel data from 28 prefecture-level cities in the UAMRYR were utilized as sample data. The data were gathered from the 2011–2020 China Urban Statistics Yearbook, the 2011–2020 Statistical Yearbooks of each province and city, and the statistical bulletin; missing data were made up using either the mean method or the interpolation method (Table 2).

**Construction of the index system.**   Under the synergy of the complex economic system, resilience and quality combine to foster the economy's synergistic and sustained evolutionary development. In this study's evaluation of the relationship between economic resilience and high-quality economic development, there are therefore two layers of system synergy: the first layer is the synergistic development of the resilient system and the high-quality economic development system, which includes sustainable economic development; the second layer is the respective synergistic system of the resilient system and the high-quality economic development system. The second layer consists of a synergistic system composed of resilient and superior economic development systems.

Briguglio et al. [39] developed a national economic resilience index evaluation system using 86 countries (regions) as research objects to measure economic resilience. Oxborrow et al. [40] investigated the economic resiliency of the Middle East through interviews and surveys. From 2006–2015, Zhang and Feng [41] examined the economic resiliency of 30 Chinese provinces using four measures, including fiscal revenue and the quantity of actual foreign capital employed. Zhang [42] devised an indicator system to compare China's and other nations' economic toughness from two perspectives: market efficiency and government efficiency.

**Table 2.  Basic data.**

| Data | Period | Source | URLs |
|---|---|---|---|
| **Panel data of 28 prefecture-level cities in the UAMRYR** | 2010–2019 | Hubei Statistical Yearbook | http://tjj.hubei.gov.cn |
| | | Hunan Statistical Yearbook | http://www.hunan.gov.cn |
| | | Jiangxi Statistical Yearbook | http://tjj.jiangxi.gov.cn |
| | | China City Statistical Yearbook | http://www.tjcn.org |
| | | Statistical Bulletin of each prefecture-level city | http://www.tjcn.org/tjgb |

**Table 3. Comprehensive evaluation index system of economic resilience.**

| Target | Guideline | Indicators | Explanation of indicators |
|---|---|---|---|
| **Level of economic resilience development** | Resistance | GDP (+) | GDP (billion yuan) |
| | | Per capita disposable income of urban residents (+) | Per capita disposable income of urban residents (yuan) |
| | | Savings balance of urban and rural residents (+) | Savings balance of urban and rural residents (billion yuan) |
| | | HHI (-) | Herfindahl- Hirschman Index |
| | | Actual level of foreign investment (-) | Actual utilization of foreign investment (billion yuan) |
| | | Urban registered unemployment rate (-) | Urban registered unemployment rate (%) |
| | Recovery | Level of fiscal self-sufficiency (+) | Local fiscal revenue / local fiscal expenditure (%) |
| | | Local fiscal expenditure (+) | Local government resource allocation capacity (billion yuan) |
| | | Total retail sales of social consumer goods (+) | Regional market size (billion yuan) |
| | Evolution | Urbanization rate (+) | Urban resident population/total resident population (%) |
| | | Intensity of financial education expenditure (+) | Local fiscal expenditure on education/local fiscal expenditure (%) |
| | | Number of patents granted per 10,000 people (+) | Number of patents granted/total resident population (items/million) |
| | | Investment in scientific research (+) | R&D expenditure/regional GDP (%) |

Note: In the table, "+" indicates that the indicator is positive, and "-" indicates that the indicator is negative, as in Table 4.

Considerations that influence economic resilience, include industrial structure, urbanization, social capital, cultural factors, etc. Drawing on the former approach, the economic resilience evaluation system outlined in this study is based on three dimensions: resistance, recovery, and evolution [43]. As demonstrated in Table 3, the development of specific indicators is based on studies conducted by Qi et al. [44] and Ding et al. [45]. The Herfindahl- Hirschman Index (HHI) assesses the structural variety of industries; the greater the HHI, the less diverse the industries. The formula is as follows.

$$HHI = \sum_{i=1}^{N} \left( \frac{X_i}{X} \right)^2 = \sum_{i=1}^{N} S_i^2$$

where $N = 3$, $X_i$, $X$, and $S_i$ represent the output value of industry I regional GDP, and the proportion of industry I output value to regional GDP, respectively.

The economy's total factor productivity is widely used to gauge the quality of high-quality economic development [46, 47]. The measurement of a single indicator, such as total factor productivity, has limits and cannot give objective and exhaustive evaluation results. Consequently, a growing number of academics are constructing a comprehensive index system for evaluating high-quality economic development. Ma et al. [48] evaluated the quality of regional high-quality economic development along five dimensions: supply, demand, efficiency, operation, and openness. Li et al. [15] devised an evaluation system that takes into account five factors: economic vitality, innovative efficiency, green development, the quality of life of the people, and social harmony. Liu et al. [49] evaluated the economic development of Northeast China from the perspective of the new development concept. As stated in Table 4, this research drew on relevant studies [17, 50] to design an evaluation index system for the high-quality economic development of the UAMRYR based on the new development concept.

**Table 4. Comprehensive evaluation index system of high-quality economic development.**

| Target | Guideline | Indicators | Explanation of indicators |
|---|---|---|---|
| **Level of high-quality economic development** | Innovative Development | Number of patents granted (+) | Total number of patents granted (items) |
| | | Investment in R&D personnel (+) | Total number of R&D personnel (10,000 people) |
| | | Fiscal expenditure intensity on science and technology (+) | Local fiscal expenditure on science and technology / local fiscal expenditure (%) |
| | Coordinated Development | Industrial structure rationalization index (-) | Industry Theil index |
| | | Industrial structure advanced index (+) | Proportion of output value of primary industry×1+Proportion of output value of secondary industry×2+Proportion of output value of tertiary industry×3 |
| | | Urban-rural income ratio (-) | Per capita disposable income of urban residents/per capita disposable income of rural residents |
| | | Urban-rural consumption ratio (-) | Per capita consumption expenditure of urban residents/per capita consumption expenditure of rural residents |
| | Green Development | Green coverage rate of built-up areas (+) | Green coverage rate of built-up areas (%) |
| | | Wastewater emissions per unit GDP (-) | Total wastewater discharge/GDP (million tons/billion yuan) |
| | | Emission of waste gas per unit of GDP (-) | Industrial $SO_2$ emissions/GDP (tons/billion yuan) |
| | | General solid waste comprehensive utilization rate (-) | General solid waste comprehensive utilization / (general solid waste generation + comprehensive utilization of storage in previous years) (%) |
| | Open Development | Degree of dependence on foreign trade (+) | Total export-import volume / regional GDP (%) |
| | | Intensity of foreign capital utilization (+) | Actual utilization of foreign investment / regional GDP (%) |
| | | Balance of trade (+) | Gross export value–gross import value (billion yuan) |
| | Inclusive Development | Number of general colleges and universities (+) | Number of general colleges and universities (%) |
| | | Number of hospital beds per 10,000 people (+) | Number of hospital beds/total resident population (sheets/million) |
| | | Number of urban basic endowment insurance participants (+) | Number of urban basic endowment insurance participants (10,000 people) |

## Results

### A comprehensive evaluation of time and space

Fig 1 illustrates the time-series features and spatial divergence of the UAMRYR's overall economic resilience and high-quality economic development. From 2010–2019, the proportion of cities with economic toughness above the medium-high value increased from 7.14–50%. Moreover, economic toughness increased dramatically. In particular, the average level of economic difficulty climbed by 78.88% between 2010 and 2015. Since the signing of the Framework Agreement on Strategic Cooperation to Accelerate the Construction of Urban Agglomerations in the Middle Reaches of the Yangtze River by Hubei, Hunan, and Jiangxi provinces in early 2012, they have led to a broad consensus in many areas, including transportation tourism, economic development, and infrastructure construction, and have continued to strengthen government-to-society cooperation. The average level of economic resilience level in 2019 was 36.30% greater than in 2015, albeit at a slower rate of expansion. In March 2015, the State Council formally approved the Development Plan for the Urban Agglomeration in the Middle Reaches of the Yangtze River, which was instrumental in enhancing the level of economic resilience and played a significant role in promoting the

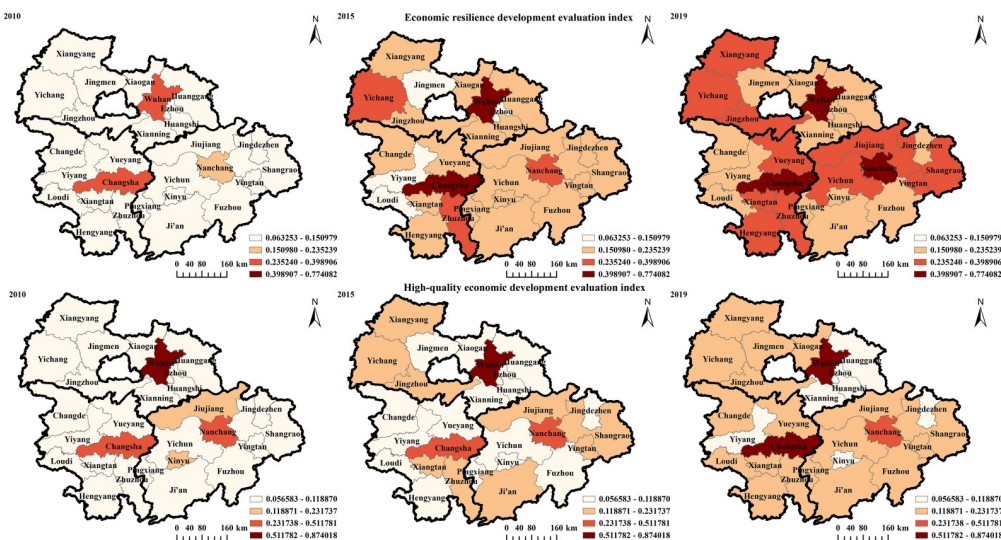

**Fig 1. Spatio-temporal evolution of economic resilience and high-quality economic development in the UAMRYR.** The image was created using ArcGIS 10.2 software with the authors' data.

overall average value to reach a medium-high level in 2017. The distribution of Wuhan, Changsha, and Nanchang spatially defined the economic robustness of the UAMRYR. This was directly tied to both the natural conditions of Jiangxi Province and the strategic measures undertaken by its government departments, such as the integration of Nanchang and Jiujiang.

From 2010–2019, in the UAMRYR, the percentage of cities in the low-value zone of economic high-quality economic development level decreased from 85.71–25%; the cities in the medium-high value and above were Wuhan, Changsha, and Nanchang; the level of economic high-quality economic development increased in general. From 2010–2015, UAMRYR's economic resilience and level of high-quality economic development rose. Despite the impact of the 2008 global financial crisis on the economic development of the provinces of Hubei, Hunan, and Jiangxi, the state's series of programs relocated investment to the central and western areas, offering additional development prospects and supporting urban economic growth. From 2015–2019, the UAMRYR maintained a sustained level of high-quality economic development. Since the 18[th] National Congress, the Hubei, Hunan, and Jiangxi provinces have been executing an innovation and rise strategy with supply-side structural reform as the core line to drive high-quality economic development. The spatial distribution of the UAMRYR's economic quality development and economic resilience was comparable to those of Wuhan, Changsha, and Nanchang. Concurrently, a great number of low-value or low-value places were located in their proximity due to the provincial capital cities' siphoning influence on the high-quality aspects of the neighboring cities, generating depressions around them. Comparatively, the UAMRYR has a lower level of economic quality development than economic resilience. High-quality economic development is measured by incorporating the economic system, environmental protection, and social security. The UAMRYR is rich in arable land, water resources, and mineral resources, which enables them to fully develop related resource-based industries. However, the UAMRYR's long dependence on resources and unreasonable industrial structure has resulted in severe pollution, and the ecological environment requires further improvement.

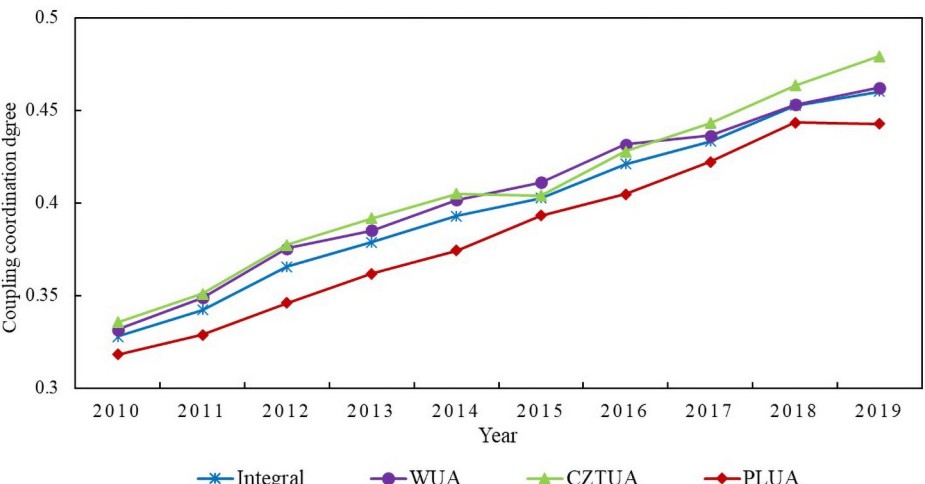

**Fig 2. Temporal evolution of the coupling coordination degree between economic resilience and high-quality economic development in the UAMRYR.** Note: WUA–Wuhan urban agglomeration, CTZUA–Chang-Zhu-Tan urban agglomeration, PLUA–Poyang Lake urban agglomeration.

## Coupling relationship analysis

**Time-series variation of the coupling relationship.** Fig 2 demonstrates the coupling coordination between the UAMRYR as a whole and the economic resilience and high-quality economic development of each sub-urban agglomeration. Chang-Zhu-Tan urban agglomeration had the fastest-growing coupling coordination level, reaching basic incoordination in 2015; Wuhan urban agglomeration's coupling coordination level was essentially synchronized with the UAMRYR as a whole, ranging from 0.33 to 0.46; and Poyang Lake urban agglomeration's coupling coordination level was relatively low, ranging from 0.31 to 0.44. The increasing degree of coupling coordination between economic resilience and high-quality economic development in the UAMRYR indicated that their synergy was improving. The overall mean value increased from 0.33 in 2010 to 0.46 in 2019, and there was still space for improvement.

The coupling coordination degree between economic resilience and economic quality development level in 28 prefecture-level cities over several years is depicted in Table 5. In 2010, only Wuhan and Changsha achieved the coordination level of linkage between economic resilience and the level of high-quality economic development in the UAMRYR. By 2015, the coupling coordination values of all 28 prefecture-level cities were greater than 0.2, indicating moderate or greater incoordination. The coupling coordination levels of 22 Chinese cities, including Wuhan, Changsha, and Nanchang, had increased. In 2019, the coupling coordination values of 28 prefecture-level cities were all larger than 0.3, signifying mild incoordination and above; Zhuzhou, Wuhan, Changsha, and Nanchang have coupling coordination levels of primary coordination and above.

**Spatial variation of the coupling relationship.** The spatial pattern concentrated on Wuhan, Changsha, and Nanchang revealed that the coupling coordination degree of economic resilience and high-quality economic development was better in the UAMRYR. As depicted in Fig 3, in 2010, the coupling coordination degree between economic resilience and high-quality economic development was most apparent in Wuhan, Changsha, and Nanchang in 2010, whereas cities with moderate and mild disturbances surrounded their peripheries. The coupling coordination degree in Wuhan, Changsha, and Nanchang was still clearly superior to that of other cities in 2015, although the number of cities with moderate problems in their

**Table 5. The coupling coordination degree between economic resilience and high-quality economic development in the UAMRYR from 2010–2019.**

| City | 2010 | 2015 | 2019 | City | 2010 | 2015 | 2019 |
|---|---|---|---|---|---|---|---|
| Wuhan | 0.636 | 0.775 | 0.906 | Yueyang | 0.295 | 0.362 | 0.437 |
| Huangshi | 0.312 | 0.376 | 0.403 | Changde | 0.298 | 0.378 | 0.422 |
| Yichang | 0.334 | 0.446 | 0.498 | Yiyang | 0.254 | 0.318 | 0.388 |
| Xiangyang | 0.339 | 0.432 | 0.490 | Loudi | 0.266 | 0.306 | 0.389 |
| Ezhou | 0.299 | 0.315 | 0.334 | Nanchang | 0.478 | 0.563 | 0.655 |
| Jingmen | 0.263 | 0.336 | 0.400 | Jingdezhen | 0.298 | 0.349 | 0.383 |
| Xiaogan | 0.272 | 0.361 | 0.411 | Pingxiang | 0.278 | 0.348 | 0.423 |
| Jingzhou | 0.310 | 0.380 | 0.431 | Jiujiang | 0.337 | 0.408 | 0.457 |
| Huanggang | 0.276 | 0.352 | 0.389 | Xinyu | 0.338 | 0.368 | 0.396 |
| Xianning | 0.270 | 0.332 | 0.354 | Yingtan | 0.314 | 0.401 | 0.431 |
| Changsha | 0.552 | 0.664 | 0.782 | Ji'an | 0.285 | 0.382 | 0.414 |
| Zhuzhou | 0.354 | 0.442 | 0.513 | Yichun | 0.260 | 0.372 | 0.433 |
| Xiangtan | 0.339 | 0.388 | 0.453 | Fuzhou | 0.274 | 0.357 | 0.402 |
| Hengyang | 0.322 | 0.371 | 0.445 | Shangrao | 0.317 | 0.381 | 0.430 |

peripheries had decreased to 20. In 2019, 65 percent of cities with moderate disorders were elevated to near disorders, and the coordination between urban economic resilience and high-quality economic development was optimized further.

The spatial development pattern within each sub-urban agglomeration was closely related to the province's development strategy, with Wuhan, Yichang, Xiangyang, and other cities in the Wuhan urban agglomeration having higher coupling coordination degree and improving more quickly, roughly following Hubei Province's regional development strategy of "one main and two vices"; Yueyang and Hengyang in the Chang-Zhu-Tan urban agglomeration have a higher of coupling coordination degree.

**Identification of factors influencing the coupling coordination.** Economic resilience and high-quality economic development are complex giant systems comprised of multiple subsystems. To investigate and evaluate the shortcomings of regional economic development in the UAMRYR, the driving force of relevant indicators acting on the coupling coordination degree, as well as its evolution process, are detected using geographic probes, and the top ten indicators ranked by influence q-value are identified, as shown in Table 6.

The explanatory power of the indicator of total retail sales of social consumer goods concerning the coupling coordination degree in the economic recovery subsystem was greater than 0.9 in 2010 and slightly decreased in 2015 and 2019. Increases in government fiscal spending between 2010 and 2015 can effectively increase resident consumption, which had a bigger impact on enhancing the coupling coordination degree. Greater resilience can spread

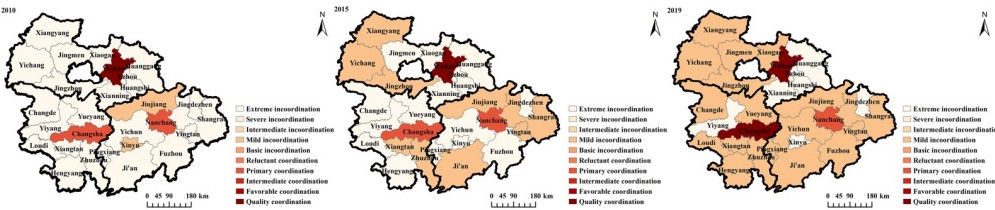

**Fig 3. Spatio-temporal evolution of the coupling coordination degree between economic resilience and high-quality economic development in the UAMRYR.** The image was created using ArcGIS 10.2 software with the authors' data.

**Table 6. Results of driving factors of the coupling coordination degree between economic resilience and high-quality economic development.**

| 2010 | | 2015 | | 2019 | |
|---|---|---|---|---|---|
| **Driving factors** | ***q*** | **Driving factors** | ***q*** | **Driving factors** | ***q*** |
| Number of urban basic endowment insurance participants | 0.917 | Investment in R&D personnel | 0.923 | Actual level of foreign investment | 0.899 |
| Total retail sales of social consumer goods | 0.903 | Number of urban basic endowment insurance participants | 0.913 | Per capita disposable income of urban residents | 0.897 |
| Industrial structure advanced index | 0.891 | Actual level of foreign investment | 0.902 | Gross Regional Product | 0.897 |
| Actual level of foreign investment | 0.875 | Total retail sales of social consumer goods | 0.890 | HHI | 0.896 |
| Number of general colleges and universities | 0.872 | Savings balance of urban and rural residents | 0.886 | Number of patents granted per 10,000 people | 0.895 |
| Investment in R&D personnel | 0.868 | Local fiscal expenditure | 0.859 | Investment in R&D personnel | 0.891 |
| Number of hospital beds per 10,000 people | 0.814 | Number of general colleges and universities | 0.840 | Number of general colleges and universities | 0.858 |
| GDP | 0.810 | Number of patents granted per 10,000 people | 0.839 | Number of urban basic endowment insurance participants | 0.857 |
| Savings balance of urban and rural residents | 0.795 | Level of fiscal self-sufficiency | 0.837 | Total retail sales of social consumer goods | 0.843 |
| Local fiscal expenditure | 0.792 | GDP | 0.831 | Urbanization Rate | 0.838 |

and transmit shock impact, improve system coupling coordination, assure the orderly promotion of high-quality economic development, and achieve sustainable economic development as demonstrated. Indicators such as GDP and the savings balance of urban and rural residents had a substantial positive driving influence on the coupling coordination degree in the economic resistance subsystem, and GDP and per capita disposable income of urban residents contributed more in 2019. As a negative indicator, the q-value of the actual level of foreign investment was generally increasing and surpassed 0.9 in 2015, indicating that since the "Belt and Road" strategy was proposed, the provinces of Hubei, Hunan, and Jiangxi had implemented a series of measures and actions in response to the national strategy, and had continuously strengthened their cooperation with countries along the route. Open development, as a significant determinant of high-quality economic development, open development had a substantial explanatory power on the coupling coordination degree between two systems. In the process of participating in cross-border capital flows, it was essential to address how to reduce external dependence on capital, boost innovation transfer in reverse, and then increase economic resilience in the context of the Double Cycle. Owing to provincial governments' execution of the rise of innovation policy in recent years, the contribution of the number of patents granted per 10,000 people to the subsystem of economic resilience evolution has been gradually growing. Innovations in science and technology were altering the input mix of components and guiding the transformation and upgrading of industries, consolidating the coupling coordination degree by maximizing the development levels of both.

In 2010, the geographically divergent explanatory power of each indicator for the coupling coordination degree between economic resilience and high-quality economic development exceeded 0.5. The indicator of the number of urban basic endowment insurance participants in the inclusive subsystem had a q-value greater than 0.90, as did the indicators of general colleges and universities, and hospital beds per 10,000 people were also greater than 0.80. Even though their q-values decreased slightly from 2015–2019, they nevertheless maintained a high level overall, indicating that endowment insurance was a significant element in determining the level of linkage coordination between economic resilience and high-quality economic development. Zou and Wang et al. [51, 52] also showed that endowment insurance fund expenditure and coverage can greatly encourage the consumption of urban inhabitants and that complete social security increases expectations. Investment in R&D personnel has a significant driving effect on the coupling coordination degree of the two systems in the innovation

subsystem, and showed a gradual increase trend, indicating that this indicator not only played a significant role in promoting the transformation and quality of economic development, but also enhanced economic resilience, so that the economic system can renew and evolve after disturbances, and improved the overall coupling coordination degree. Comparatively, the coordination, green, and open subsystems contribute relatively little to the coupling coordination degree in 2010, except for the industrial advanced indicator in the coordination subsystem. Interaction detection revealed that the detection factors in 2010, 2015, and 2019 exhibited a double-factor enhancement or nonlinear enhancement after interaction, indicating that the formation of the spatio-temporal divergence pattern of coupled and coordinated economic resilience and high-quality economic development in the UAMRYR was due to multi-factor interaction and synergy. As a result, the author believes that the construction of a composite development strategy in the collaborative development of the UAMRYR can assist in enhancing the level of coupling coordination between economic resilience and high-quality economic development, thereby boosting the level of regional sustainable development. In addition to exchange and cooperation, an additional opening is required.

## Discussion

### Coupling coordination mechanism

As seen in Fig 4, economic resilience is the basis and precondition for high-quality economic development, and high-quality economic development is the outcome and performance of economic resilience. Enhanced economic resilience will, on the one hand, foster high-quality urban economic development. As a subsystem of the economic development system, the economic resilience system played a crucial supporting role. The improvement of economic resistance, recovery, and evolution will optimize the industrial structure, science, and technology innovation development, decreasing pollution emissions, promoting high-quality economic

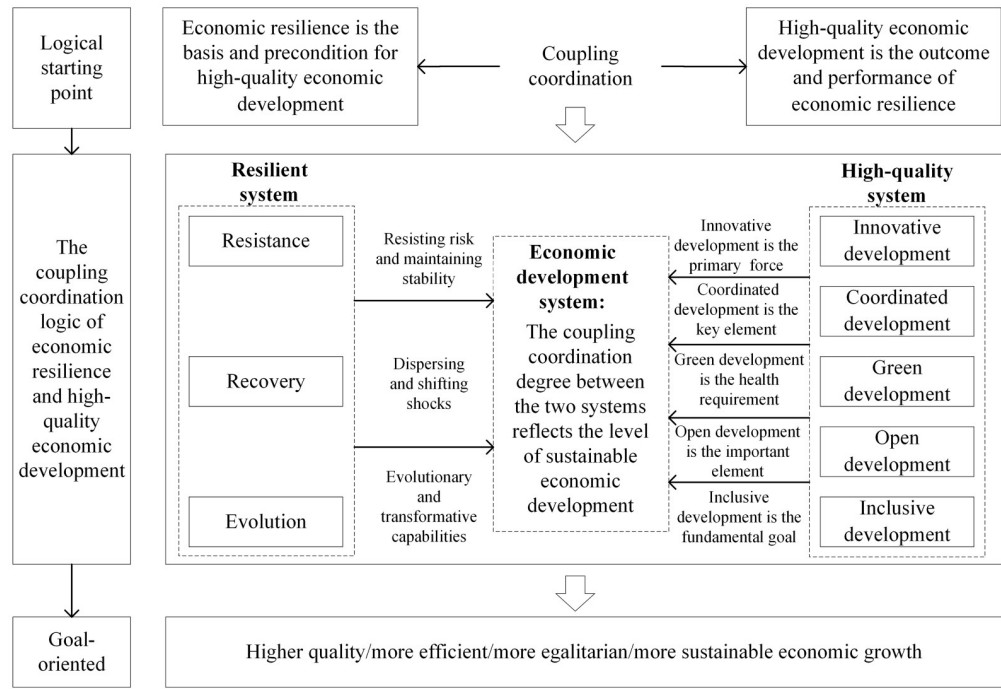

**Fig 4. Synergistics system of economic resilience and high-quality economic development.**

development, and speeding the sustainable growth of the economic system. The number of patents granted per 10,000 people in Wuhan, for example, increased from 12.14 in 2010 to 43.33 in 2019, and the upgrade of the evolution of economic resilience represents a series of changes in industrial structure and ecological environment, which has had a profound effect here on the transition from primary to high-quality coordinated development. The number of patents granted per 10,000 people in Ezhou decreased from 14.64 in 2010 to 6.64 in 2019, showing that the coupling coordination degree in 2019 remains mildly uncoordinated.

On the other hand, economic resilience will be bolstered by high-quality economic development. To achieve good development in an environment of innovation, coordination, green, openness, and inclusiveness, the economic system's sustainability is bolstered; this sustainability also affords the chance to transform and strengthen the resilient system, thereby significantly enhancing economic resilience In addition, if medical health and education degrees were assured, it would be easier to unleash the consumer potential, encourage the steady forward movement of the social economy, achieve quality economic growth, and optimize and enhance the economic resilience system. In Wuhan, a sub-provincial city, the number of urban basic endowment insurance participants climbed from 3,095,800 in 2010 to 4,823,000 in 2019, surpassing 27 other cities with a more comprehensive inclusive subsystem and citizens with a greater consumption capacity. Wuhan's coupling coordination was significantly greater than Yingtan's because general prefecture-level cities like Yingtan had fewer participants in urban basic endowment insurance than the other 27 cities.

This indicates that the linking mechanism exists not only in provincial capitals but also in prefecture-level cities. Economic resilience and high-quality economic development were mutually reinforcing, creating a powerful incentive for the sustainable development of urban economic systems to achieve higher quality, more efficient, more egalitarian, and more sustainable economic growth.

## Strategies to promote the coupling coordination degree

Several studies have examined ways to quantify the connection between economic resilience and high-quality development. Using multiple panel regression models for the western region, Sun and Yuan [28] demonstrate that economic resilience contributes significantly to high-quality economic development in the short term, whereas the two have a causal relationship in the long term. Guan and Zhang [29] employ panel regression models to demonstrate how economic resilience might contribute to the Yangtze River Delta's high-quality economic development. Most of the aforementioned research examines the impact of economic resilience in supporting high-quality development by employing panel regression models and focusing on the examination of the coupling coordination process and the current condition. Lack of driving effect for coupling coordination mechanism employing techniques such as geographical detector to measure indicator factors for each subsystem of economic resilience and high-quality economic growth. Consequently, this study presents the following three significant contributions: (1) Using the UAMRYR as a research object improves the study of urban agglomeration in central China. (2) To quantify the extent of coupling and coordination between economic resilience and high-quality economic development in the UAMRYR. (3) Combining the CCDM model and GD to precisely measure the impact values of economic resilience and economic quality development index variables to statistically examine their coupling coordination mechanism. Based on these findings, this study concludes that the coupling coordination between economic resilience and high-quality economic development in the UAMRYR exhibits an upward trend, but is still dominated by a state of incoordination and has substantial development potential. Moreover, due to the lack of research on the coupling

coordinated driving mechanism in the existing studies, this study uses GD to quantify each indicator and discovers that the total retail sales of social consumer goods, the number of urban basic endowment insurance participants, the number of general colleges and universities, and the investment in R&D personnel are the most influential factors on the coupling coordination degree. The study's results provide more focused recommendations for the economic development of the UAMRYR and useful references for the sustainable regional development of other emerging nations.

Certain advances in the development of the UARMRYR have happened since the 13th Five-Year Plan, and provincial governmental cooperation mechanisms have been formed, but the overall trend of collaboration is poor, resulting in slower economic growth. To effectively promote the high-quality economic growth of the UAMRYR, the development strategy must be adapted to the real situation. The subsequent policy consequences are proposed:

Firstly, implement the Strong Provincial Capitals plan and fortify the Growth Poles. The Hunan Province Twelfth Party Congress suggested pursuing the "strong provincial capital" policy, which is essential to Changsha's economic growth. Wuhan, Changsha, and Nanchang should lead UAMRYR to enhance additional linkages with adjacent cities, actualize the growth of the same city, and promote optimal factor combination.

Secondly, implement the Integration Strategy and construct a Synergistic Network. We will implement the 14th Five-Year Strategy for the implementation of the UAMRYR following the central government's top-level design requirements. The geographical plan of "three cores, three rings, three belts, and numerous nodes" is intended to promote the coordinated development of the small, medium, and large cities and small towns as well as decrease spatial disparities within the city circle. It is led by a scientific labor division and synergistic growth. Simultaneously, it will speed the urban agglomeration's sub-center city development to achieve integrated development as quickly as feasible.

Finally, improve Resilience to achieve Excellent Quality. To support the high-quality growth of the economy and to increase its resilience in the three areas of resistance, recovery, and evolution. Promote the collaboration of the Optical Valley Technology Innovation Corridor, the Xiang River West Coast Science and Technology Innovation Corridor, and the Gan River Cross-Strait Science and Technology Innovation Corridor, as well as the industrial synergy and innovation of the three metropolitan areas. Using the Multi-Combination development approach and enhancing resource integration, we want to achieve breakthroughs in critical areas such as technological innovation and industrial optimization, and to establish a collaborative innovation demonstration area.

## Conclusion

The coupling coordination between economic resilience and high-quality economic development is a dynamic process. To quantitively evaluate the dynamic change in the coupling coordination degree between economic resilience and high-quality economic development in the study area, the aggregated index system was introduced and the method combining CCDM with GD was developed in response to a changing environment. The following are the study's conclusions:

1. The economic resilience and level of high-quality economic development of the UAMRYR have increased, while the level of high-quality economic development lags behind economic resilience. Wuhan, Changsha, and Nanchang have a significant siphoning effect on nearby cities due to their spatial economic resilience and level of high-quality economic development.

2. The coupling coordination degree between economic resilience and high-quality economic development is increasing, with the overall average value increasing from 0.33 in 2010 to 0.46 in 2019, and the evolution of each sub-urban agglomeration is essentially consistent with it, which is on the verge of disorder and has significant development potential. Spatial analysis of the UAMRYR coupling coordination reveals a "polycentric" spatial distribution pattern, with Wuhan, Changsha, and Nanchang as the core, and a huge number of dysfunctional cities dispersed around its periphery.

3. The economic resilience resistance subsystem has a significant impact on the coupling coordination degree. High-quality economic development of the five major subsystems, creative and inclusive subsystems drive coupling coordination degree synthetically, whereas coordinated, green, and open subsystems are insignificant. The total retail sales of consumer goods, the number of urban basic endowment insurance participants, the number of general colleges and universities, and the investment in R&D personnel are the most influential factors on coupling coordination degree, and their explanatory power for the coupling coordination degree is greater than 0.8. Each element's interaction shows a complementary enhancement impact and a double-factor enhancement effect, and each factor grows synergistically to produce sustainability.

4. Economic resilience and quality economic development are complementary and cascading, providing a powerful drive for the city's economic system's sustainable development. To encourage the accelerated development of the UAMRYR, the focused promotion of paired coordination of urban economic resilience and high-quality economic development has become an inevitable choice.

## Author Contributions

**Writing – original draft:** Shenglan Ma.

**Writing – review & editing:** Junlin Huang.

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
