## [Decision Letter · Decision Letter 0]

17 Oct 2022

PONE-D-22-22780Research on the spatial-temporal coupling mechanism of economic resilience and high-quality economic development in the urban agglomeration in the middle reaches of the Yangtze RiverPLOS ONE

Dear Dr. HUANG,

Thank you for submitting your manuscript to PLOS ONE. After careful consideration, we feel that it has merit but does not fully meet PLOS ONE’s publication criteria as it currently stands. Therefore, we invite you to submit a revised version of the manuscript that addresses the points raised during the review process. Please resumbit your revised manuscript by Nov 10, 2022

We look forward to receiving your revised manuscript.

Kind regards,

Xin Shen

Academic Editor

PLOS ONE

Journal Requirements:

3. We note that Figures 1 and 3 in your submission contain [map/satellite] images which may be copyrighted. All PLOS content is published under the Creative Commons Attribution License (CC BY 4.0), which means that the manuscript, images, and Supporting Information files will be freely available online, and any third party is permitted to access, download, copy, distribute, and use these materials in any way, even commercially, with proper attribution. For these reasons, we cannot publish previously copyrighted maps or satellite images created using proprietary data, such as Google software (Google Maps, Street View, and Earth). For more information, see our copyright guidelines: http://journals.plos.org/plosone/s/licenses-and-copyright.

a. You may seek permission from the original copyright holder of Figures 1 and 3 to publish the content specifically under the CC BY 4.0 license.  

4. We note you have included a table to which you do not refer in the text of your manuscript. Please ensure that you refer to Table 1 in your text; if accepted, production will need this reference to link the reader to the Table.

Reviewers' comments:

Reviewer's Responses to Questions

**Comments to the Author**

1. Is the manuscript technically sound, and do the data support the conclusions?

Reviewer #1: Partly

Reviewer #2: Yes

2. Has the statistical analysis been performed appropriately and rigorously? 

Reviewer #1: Yes

Reviewer #2: Yes

3. Have the authors made all data underlying the findings in their manuscript fully available?

Reviewer #1: Yes

Reviewer #2: Yes

4. Is the manuscript presented in an intelligible fashion and written in standard English?

Reviewer #1: Yes

Reviewer #2: No

5. Review Comments to the Author

Reviewer #1: In this study, the electricity-related water network embodied in power system has been constructed, the properties based on the multi-regional input-output model and complex network analysis has also been done.Before the further consideration can be made, several issues in the manuscript should be addressed as follows:

1. While the author presents the Abstract, answer the questions carefully: What problem did you study and why is it important? What methods did you use? What were your main results? And what conclusions can you draw from your results? Please make your abstract with more specific and quantitative results while it suits broader audiences. Frankly, the current background is too long to follow, it is suggested that author can refine this part. Go straight to introduce the method and the most important data-driven findins of your own study may be much better.

2. The current Introduction should be further improved. A good one includes at least four aspects: motivation/background, literature review, aim and contribution, and organization of the remains of the study. Avoiding to put massive bibliographies behind one sentence. Such as XXXXX [1-5], OR 1, 2, 3, 4, 5; all references should be cited with detailed and specific descriptions.

3. Literature Review has the chance to be further improved: it seems that the authors have made the retrospection. However, via the review, what issues should be addressed? What is the current specific knowledge gap? What implication can be referred? The above questions should be point-by-point answered clearly. This work discussed the economic resilience and high-quality economic development in the urban agglomeration. At the demand side, buildings show the most significant potential in cost-effectiove emission reduction, which is worthy to be discussed. Here, some latest literature investigating the low carbon roadmap or carbon neutral pathway of building sector (especially the carbon emission mitigation in building operations) is worhy to added and discussed. E.g., DOI: 10.1016/j.resconrec.2022.106481 DOI: 10.1016/j.apenergy.2022.119401 DOI: 10.1016/j.scitotenv.2022.157679

4. For some details, be sparing in the use of tables and ensure that the data presented in them do not duplicate results described elsewhere in the article. It is suggest to avoid using vertical rules and shading in table cells. It is also suggested that the key findings should be summarized in Conclusion one by one with the marks of 1. 2. 3. or i. ii. iii. etc.

5. As a key part of a paper, Discussion should show the readers at least two elements: "breadth" and "depth". "Breadth" reflects whether the analytical results can be explained via different approaches. "Depth" reflects whether the analytical results completely answer the questions raised in Introduction. My first sense shows the current Discussion is without enough insight. This should explore the significance of the results of the work, not repeat them. A combined CONCLUSION and Discussion section is not allowed. Besides, avoid extensive citations and discussion of published literature.

Reviewer #2: The paper titled "Research on the spatial-temporal coupling mechanism of economic resilience and high-quality economic development in the urban agglomeration in the middle reaches of the Yangtze River" . Overall, the paper's quality can be improved; however, the following major revisions can be considered:

(1) Abstract needs to be rewritten well. It is necessary to highlight the study's main findings without mentioning that the policy and limits are suggested at the end of the article. Elaborate abstract in more detail in terms of research methodology, including data duration, and specific policy implication.

(2) Figure 1 and Figure 3 are blurred and indistinct. It is suggested to make the drawing again.

(3) Many punctuation marks do not have Spaces between words. Please use proper punctuation.

(4) The language needs further polishing, and it is best to invite a native English speaker of relevant academic research to help with the revision.

(5) I recommend authors to explicitly specify the novelty of this research paper?

(6) The introduction needs to be expanded further to cover all the critical aspects of the topic. I suggest authors to consider the following articles for citations, it will strengthen the quality of this draft:

doi.org/10.3390/su14074212

doi:10.3390/ijerph16173076

doi.org/10.3390/ijerph18115697

doi.org/10.3390/ijerph18179105

doi.org/10.3390/ijerph19020958

6. PLOS authors have the option to publish the peer review history of their article (what does this mean?). If published, this will include your full peer review and any attached files.

Reviewer #1: No

Reviewer #2: No

---

## [Author Response · Author response to Decision Letter 0]

9 Nov 2022

Response to the Editor and Reviewers

Manuscript No.: PONE-D-22-22780

Title: Analysis of the spatio-temporal coupling coordination mechanism supporting economic resilience and high-quality economic development in the urban agglomeration in the middle reaches of the Yangtze River

Authors: Shenglan Ma, Junlin Huang (corresponding author)

E-mail: huangjl1020@163.com

Dear Prof. Shen:

We would like to thank you for your efforts in reviewing and handling our manuscript. We have addressed each of the concerns of the editor, the reviewer #1 and the reviewer #2, and revised the manuscript according to their valuable suggestions. Our responses to the comments and the modifications that we have made are listed below. We believe the quality of the manuscript has been improved and hope that you will find it acceptable for publication.

Thank you very much for your consideration. We are looking forward to hearing from you as soon as possible

Yours sincerely,

Junlin Huang

School of Geographical Sciences, Hunan Normal University

36# Lushan Road of Changsha, Hunan, China 410081

Replies to Editor

Comments: 

2) In your Data Availability statement, you have not specified where the minimal data set underlying the results described in your manuscript can be found.

3) We note that Figures 1 and 3 in your submission contain [map/satellite] images which may be copyrighted.

4) We note you have included a table to which you do not refer in the text of your manuscript.

Reply: Thank you very much for your insightful comments, which further help use in improvement of quality of the manuscript. The following is a point-by-point response to your concerns.

Comments: 1) Please ensure that your manuscript meets PLOS ONE's style requirements, including those for file naming.

Reply: Thank you for your suggestion. In the revised manuscript, we have revised the style of the manuscript in accordance with the PLOS ONE's style requirements.

Comments: 2) In your Data Availability statement, you have not specified where the minimal data set underlying the results described in your manuscript can be found.

Reply: Thank you for this suggestion. We have added the Supporting Information. Moreover, the source of all basic data is presented in Table 2 of the revised manuscript, and we have provided the relevant URLs. The revised part can be found on line 260. 

Comments: 3) We note that Figures 1 and 3 in your submission contain [map/satellite] images which may be copyrighted.

Reply: Thank you for your suggestion. We want to clarify that Figures 1 and 3 in this article are produced by us using ArcGIS 10.2, and therefore, there is no copyright conflict. Moreover, we have added “The image was created using ArcGIS 10.2 software with the authors' data” in lines 406 and 483 of the revised manuscript.

Comments: 4) We note you have included a table to which you do not refer in the text of your manuscript.

Reply: Thank you for your suggestion. We have referred to this table in the revised manuscript.

Replies to Reviewer 1

Reviewer #1:

Comments: In this study, the electricity-related water network embodied in power system has been constructed, the properties based on the multi-regional input-output model and complex network analysis has also been done. Before the further consideration can be made, several issues in the manuscript should be addressed as follows:

1. While the author presents the Abstract, answer the questions carefully: What problem did you study and why is it important? What methods did you use? What were your main results? And what conclusions can you draw from your results? Please make your abstract with more specific and quantitative results while it suits broader audiences. Frankly, the current background is too long to follow, it is suggested that author can refine this part. Go straight to introduce the method and the most important data-driven findings of your own study may be much better.

2. The current Introduction should be further improved. A good one includes at least four aspects: motivation/background, literature review, aim and contribution, and organization of the remains of the study. Avoiding to put massive bibliographies behind one sentence. Such as XXXXX [1-5], OR 1, 2, 3, 4, 5; all references should be cited with detailed and specific descriptions.

3. Literature Review has the chance to be further improved: it seems that the authors have made the retrospection. However, via the review, what issues should be addressed? What is the current specific knowledge gap? What implication can be referred? The above questions should be point-by-point answered clearly. This work discussed the economic resilience and high-quality economic development in the urban agglomeration. At the demand side, buildings show the most significant potential in cost-effective emission reduction, which is worthy to be discussed. Here, some latest literature investigating the low carbon roadmap or carbon neutral pathway of building sector (especially the carbon emission mitigation in building operations) is worthy to added and discussed. E.g., DOI: 10.1016/j.resconrec.2022.106481 DOI: 10.1016/j.apenergy.2022.119401 DOI: 10.1016/j.scitotenv.2022.157679

4. For some details, be sparing in the use of tables and ensure that the data presented in them do not duplicate results described elsewhere in the article. It is suggest to avoid using vertical rules and shading in table cells. It is also suggested that the key findings should be summarized in Conclusion one by one with the marks of 1. 2. 3. or i. ii. iii. etc.

5. As a key part of a paper, Discussion should show the readers at least two elements: "breadth" and "depth". "Breadth" reflects whether the analytical results can be explained via different approaches. "Depth" reflects whether the analytical results completely answer the questions raised in Introduction. My first sense shows the current Discussion is without enough insight. This should explore the significance of the results of the work, not repeat them. A combined CONCLUSION and Discussion section is not allowed. Besides, avoid extensive citations and discussion of published literature.

Reply: Thank you very much for your valuable comments and suggestions. We have read all comments carefully and revised the previous manuscript as suggested.

Comments: 1. While the author presents the Abstract, answer the questions carefully: What problem did you study and why is it important? What methods did you use? What were your main results? And what conclusions can you draw from your results? Please make your abstract with more specific and quantitative results while it suits broader audiences. Frankly, the current background is too long to follow, it is suggested that author can refine this part. Go straight to introduce the method and the most important data-driven findings of your own study may be much better.

Reply: Thank you for your careful reading of our manuscript and giving those suggestions. In the revised manuscript, we have carefully rewritten the abstract. We have answered the questions you mentioned in the abstract. 

(1) The problem we study is the mechanism of coupling coordination between economic resilience and high-quality economic development. The coupling coordination mechanism between economic resilience and high-quality economic development is clarified not only to improve the capacity of economies to withstand external shocks but also to boost the level of sustainable economic development. 

(2) The methods we use are the entropy method, coupling coordination model, geographical detector. This study constructs an economic resilience indicator system based on the data of 28 prefecture-level cities in the urban agglomeration in the middle reaches of the Yangtze River (UAMRYR) from 2010–2019. The entropy approach was then coupled with the coupling coordination model to determine the level of coupling coordination between the two. Finally, the geographical detector was employed to reveal the coupling coordination process and driving elements. 

(3) Our main results: Firstly, the economic resilience and high-quality economic development of the UAMRYR are on the rise; Secondly, the coupling coordination degree between economic resilience and high-quality economic development is improving, with the average value increasing from 0.33 in 2010 to 0.46 in 2019, exhibiting a Core-Periphery spatial distribution pattern with Wuhan, Changsha, and Nanchang as the core; Thirdly, the economic resilience resistance subsystem plays a substantial role in the driving influence of the coupling coordination degree, where the total retail sales of consumer goods indicator have an explanatory power of 0.8 or higher for coupled coordination. Innovation and inclusiveness thoroughly drive the coupling coordination degree among the five subsystems of high-quality economic development, where the q-value of the urban basic endowment insurance participation index is at least 0.9. Additionally, the interplay of the components demonstrates a complementary enhancement impact and a two-factor enhancement effect; Finally, economic resilience and high-quality economic development are complementary and provide a powerful incentive for the sustainable development of urban economic systems to achieve higher quality, more efficient, more egalitarian, and more sustainable economic development.

(4) Conclusions we can draw from our results: This study improves the analysis of mechanism between economic resilience and high-quality economic development, and it provides more targeted guidance for economic development in the UARMRYR. For details, see lines 23-61 in the revised manuscript.

Comments: 2. The current Introduction should be further improved. A good one includes at least four aspects: motivation/background, literature review, aim and contribution, and organization of the remains of the study. Avoiding to put massive bibliographies behind one sentence. Such as XXXXX [1-5], OR 1, 2, 3, 4, 5; all references should be cited with detailed and specific descriptions.

Reply: Thank you very much for your insightful advises. We have carefully rewritten the introduction in the revised manuscript. The introduction includes the four aspects you mentioned.

(1) Background: The 19th Communist Party of China National Congress declared that China's economy has entered the stage of high-quality economic development, and the 2018 Central Economic Work Conference proposed specific requirements for achieving high-quality economic development, indicating China’s direction toward high-quality economic development. Concurrently, the globe is becoming increasingly insecure and uncertain. However, the relationship between economic resilience and high-quality economic growth is in its infancy, and additional research is necessary to discover how economic resilience can contribute to high-quality economic development. Therefore, researching the relationship between “Resilience” and “High-Quality Development” is of considerable practical importance in the current complicated and ever-changing international and local context.

(2) Literature review: Since 1973, when Holling [1] introduced the notion of "resilience" from engineering mechanics to the study of ecosystem restoration, the concept’s content and scope have been developing. Reggiani A and Graaff T D [2] introduced the notion of resilience to the study of spatial economics for the first time; since then, resilience has been utilized in the field of economics. Academics have not yet achieved a consensus over the meaning of economic resilience. From a research standpoint, economic resilience can be subdivided into engineering resilience, ecological resilience, and evolutionary resilience; from a research methodology standpoint, it can be separated into unidimensional indicators and multidimensional indicators. Some researchers use regional sensitive index to measure economic resilience [3,4]. Others, like Martin [5], define economic resilience as resistance, recovery, re-orientation and renewal. Economic systems are characterized by disequilibrium, dynamic evolution, and complicated adaptation; hence, it is currently more acceptable to define economic resilience in various dimensions. This study focuses on economic resilience, which is the capacity of the economic system to strengthen its capacity to withstand external shocks. Therefore, economic resilience is defined as the resistance of the regional economic system to preserve its stability after a shock, the recovery to return to the original growth pattern, and the evolution to adapt and develop based on its resources. Economic systems are characterized by disequilibrium, dynamic evolution, and complicated adaptation; hence, it is currently more acceptable to define economic resilience in various dimensions. This study focuses on economic resilience, which highlights the capacity of an economic system to increase its resistance to external shocks. Consequently, economic resilience is described as the ability of a regional economic system to sustain its stability following a shock, to revert to its original growth trajectory, and to evolve and develop by relying on its endowments. Martin [5] proposed a comprehensive analysis framework of regional economic resilience influencing factors in the fields of industrial structure, labor force, finance, and institutions, and Lagravinese, Di and other researchers [6,7] conducted empirical studies in terms of industrial specialization level and policy system.

High-quality development is also known as high-quality economic development, as it focuses on the economic area. High-quality economic development highlights its systemic and comprehensive nature, which necessitates not only more efficient economic development and a more rational industrial structure but also better standards for green and low-carbon development [8-10]. Three categories comprise the discussion on the connotation of high-quality economic development. The first category is based on the New Concept for Development perspective and the primary social conflicts [11,12]. The second category emphasizes the shift of the economy from high speed to high quality [13,14]. It is based on high-quality development. The third category distinguishes between the differing requirements of narrow and broad or micro and macro [15]. Regarding the measurement of the level of high-quality economic development, some scholars utilize total factor productivity to evaluate the amount of high-quality economic development [16], but the majority of studies establish a comprehensive system of indicators [17,18]. The New Concept for Development [19,20] is the primary source for the analysis of the influencing elements, but several researchers also examine the impact of other factors on the quality of economic development from a variety of perspectives. Tang, Zhang, and other experts [21,22] suggest that enhancing the quality of urbanization can significantly boost high-quality economic growth. Xiao, Kong, and colleagues [23] identify the influence of innovation-driven strategy on high-quality development. Domestic and international research on the mechanisms of economic resilience and high-quality economic development must be expanded. According to Chen and Liang [24], economic resilience is the basis for China's economy to remain within a sustainable operating range and define the country’s continuous growth. Wang and other researchers [25,26] suggest that it is essential to consistently promote high-quality economic transformation and upgrading through bolstering economic resilience. Li [27] argues that establishing a robust and dynamic capital market can contribute to the promotion of high-quality development. Sun and Yuan [28] use the western area to demonstrate that economic resilience strongly contributes to high-quality economic development in the near term, whereas the two have a long-term causal link. Guan and Zhang [29] find that economic resilience can contribute to high-quality economic development in the Yangtze River Delta region.

(3) Contribution: Urban agglomeration in the middle reaches of the Yangtze River (UAMRYR) is an economic growth pole for the realization of the Central Rise, and the development of UAMRYR will significantly support the development the China [30]. It is crucial to examine the coupling mechanism between its economic resilience and high-quality development to enhance the region’s sustainable development [31]. Consequently, based on the data of 28 prefecture-level cities in the UAMRYR from 2010–2019, this study uses the entropy method, coupling coordination model, and geographic detector to analyze the coupling coordination relationship between the two, to provide a reference for the path selection and policy formulation of high-quality economic development in the UAMRYR, and to supplement the research on the economic development of urban agglomeration in the central region.

(4) Organization of the remains of the study: Existing research on economic resilience and high-quality economic development has made some progress, but the following limitations remain: on the one hand, in terms of research perspectives, most existing studies are limited to the one-way correlation between high-quality development and economic resilience or economic resilience to high-quality development, but there are few studies on the coupling and coordination of the synergistic interaction and two-way feedbacks between the two; on the other hand, the research content focuses on the measurement indicators or influencing factors of economic resilience or high-quality development, but the issues of how to promote the synergistic and balanced development of both and the drivers of economic resilience and high-quality economic development have not been fully demonstrated.

Moreover, in the introduction, we have adjusted the citation of references. All references have been cited with detailed and specific descriptions. For details, see lines 67-185 in the revised manuscript.

Comments: 3. Literature Review has the chance to be further improved: it seems that the authors have made the retrospection. However, via the review, what issues should be addressed? What is the current specific knowledge gap? What implication can be referred? The above questions should be point-by-point answered clearly. This work discussed the economic resilience and high-quality economic development in the urban agglomeration. At the demand side, buildings show the most significant potential in cost-effective emission reduction, which is worthy to be discussed. Here, some latest literature investigating the low carbon roadmap or carbon neutral pathway of building sector (especially the carbon emission mitigation in building operations) is worthy to added and discussed. E.g., DOI: 10.1016/j.resconrec.2022.106481 DOI: 10.1016/j.apenergy.2022.119401 DOI: 10.1016/j.scitotenv.2022.157679

Reply: We appreciate this valuable comment. We have elaborated the literature review in more detail and have presented the points you mentioned in the revised manuscript. 

(1) Problems that should be solved: It is crucial to examine the coupling mechanism between economic resilience and high-quality development.

(2) Current specific knowledge gap: Existing research on economic resilience and high-quality economic development has made some progress, but the following limitations remain: on the one hand, in terms of research perspectives, most existing studies are limited to the one-way correlation between high-quality development and economic resilience or economic resilience to high-quality development, but there are few studies on the coupling and coordination of the synergistic interaction and two-way feedbacks between the two; on the other hand, the research content focuses on the measurement indicators or influencing factors of economic resilience or high-quality development, but the issues of how to promote the synergistic and balanced development of both and the drivers of economic resilience and high-quality economic development have not been fully demonstrated.

(3) Implication: Urban agglomeration in the middle reaches of the Yangtze River (UAMRYR) is an economic growth pole for the realization of the Central Rise, and the development of UAMRYR will significantly support the development the China [30]. It is crucial to examine the coupling mechanism between its economic resilience and high-quality development to enhance the region’s sustainable development [31]. Consequently, based on the data of 28 prefecture-level cities in the UAMRYR from 2010–2019, this study uses the entropy method, coupling coordination model, and geographic detector to analyze the coupling coordination relationship between the two, to provide a reference for the path selection and policy formulation of high-quality economic development in the UAMRYR, and to supplement the research on the economic development of urban agglomeration in the central region.

In addition, we have added the mentioned literatures in the revised manuscript. (Ref. 8, lines 843-845; Ref. 9, lines 846-848; Ref.10, lines 849-851).

Comments: 4. For some details, be sparing in the use of tables and ensure that the data presented in them do not duplicate results described elsewhere in the article. It is suggest to avoid using vertical rules and shading in table cells. It is also suggested that the key findings should be summarized in Conclusion one by one with the marks of 1. 2. 3. or i. ii. iii. etc.

Reply: We thank for the reviewer’s suggestion. Firstly, we have adjusted the table in the revised manuscript without using vertical rules and shading in table cells. And in the revised manuscript, the data in the tables have not duplicated the results described in the article. Secondly, in the conclusion, we have summarized the study findings line by line with serial numbers, looking specifically at lines763-798 in the revised manuscript.

Comments: 5. As a key part of a paper, Discussion should show the readers at least two elements: "breadth" and "depth". "Breadth" reflects whether the analytical results can be explained via different approaches. "Depth" reflects whether the analytical results completely answer the questions raised in Introduction. My first sense shows the current Discussion is without enough insight. This should explore the significance of the results of the work, not repeat them. A combined CONCLUSION and Discussion section is not allowed. Besides, avoid extensive citations and discussion of published literature.

Reply: We sincerely thank the reviewer for careful reading. We have separated the discussion and the conclusion sections in the revised manuscript. The discussion has two sections. In section 1, we discuss the mechanism of coupling coordination between economic resilience and high-quality economic development. Enhanced economic resilience will, on the one hand, foster high-quality urban economic development. On the other hand, economic resilience will be bolstered by high-quality economic development. In section 2, we summarize the findings of the study, highlight the innovative points of this study and propose a development strategy. This study presents the following three significant contributions: (1) Using the UAMRYR as a research object improves the study of urban agglomeration in central China. (2) To quantify the extent of coupling and coordination between economic resilience and high-quality economic development in the UAMRYR. (3) Combining the CCDM model and GD to precisely measure the impact values of economic resilience and economic quality development index variables to statistically examine their coupling coordination mechanism. Based on these findings, this study concludes that the coupling coordination between economic resilience and high-quality economic development in the UAMRYR exhibits an upward trend, but is still dominated by a state of incoordination and has substantial development potential. Moreover, due to the lack of research on the coupling coordinated driving mechanism in the existing studies, this study uses GD to quantify each indicator and discovers that the total retail sales of social consumer goods, the number of urban basic endowment insurance participants, the number of general colleges and universities, and the investment in R&D personnel are the most influential factors on the coupling coordination degree. To effectively promote the high-quality economic growth of the UAMRYR, the development strategy must be adapted to the real situation. The subsequent policy consequences are proposed: Firstly, implement the Strong Provincial Capitals plan and fortify the Growth Poles; Secondly, implement the Integration Strategy and construct a Synergistic Network; Finally, improve Resilience to achieve Excellent Quality. For details, see lines 590-761 in the revised manuscript.

In the conclusion, we have summarized the study findings line by line with serial numbers, looking specifically at lines 763-798 in the revised manuscript.

Replies to Reviewer 2

Reviewer #2:

Comments: The paper titled "Research on the spatial-temporal coupling mechanism of economic resilience and high-quality economic development in the urban agglomeration in the middle reaches of the Yangtze River". Overall, the paper's quality can be improved; however, the following major revisions can be considered:

1. Abstract needs to be rewritten well. It is necessary to highlight the study's main findings without mentioning that the policy and limits are suggested at the end of the article. Elaborate abstract in more detail in terms of research methodology, including data duration, and specific policy implication.

2. Figure 1 and Figure 3 are blurred and indistinct. It is suggested to make the drawing again.

3. Many punctuation marks do not have Spaces between words. Please use proper punctuation.

4. The language needs further polishing, and it is best to invite a native English speaker of relevant academic research to help with the revision.

5. I recommend authors to explicitly specify the novelty of this research paper?

6. The introduction needs to be expanded further to cover all the critical aspects of the topic. I suggest authors to consider the following articles for citations, it will strengthen the quality of this draft:

doi.org/10.3390/su14074212

doi:10.3390/ijerph16173076

doi.org/10.3390/ijerph18115697

doi.org/10.3390/ijerph18179105

doi.org/10.3390/ijerph19020958

Reply: We are grateful to you for your in-depth reviews and for providing very valuable suggestions for us to improve our paper. The following is a point-by-point response to your concerns.

Comments: 1. Abstract needs to be rewritten well. It is necessary to highlight the study's main findings without mentioning that the policy and limits are suggested at the end of the article. Elaborate abstract in more detail in terms of research methodology, including data duration, and specific policy implication. 

Reply: Thank you for your careful reading of our manuscript and giving those instructive suggestions. In the revised manuscript, we have carefully rewritten the abstract. We have specified the research methodology and highlighted the main findings. The revised part can be found between lines 23-61.

Comments: 2. Figure 1 and Figure 3 are blurred and indistinct. It is suggested to make the drawing again.

Reply: We thank the reviewer for this important comment and agree that Figure 1 and Figure 3 are not clear. We have replaced the corresponding pictures with higher quality images.

Comments:3. Many punctuation marks do not have Spaces between words. Please use proper punctuation.

Reply: We are grateful to you for your careful reviews. We have corrected the punctuation in the revised manuscript.

Comments:4. The language needs further polishing, and it is best to invite a native English speaker of relevant academic research to help with the revision.

Reply: Thank you for your insightful comments on the manuscript. According to your suggestion, Editage has been invited for assistance in revising the English language, grammar, and improve clarity. Below is the certificate of Editage Language Editing Services, and relevant modifications have been made in the revised manuscript. Please check it. Thank you!

Comments:5. I recommend authors to explicitly specify the novelty of this research paper?

Reply: Thank you for your careful review. We have elaborated the novelty of the paper in the discussion. Several studies have examined ways to quantify the connection between economic resilience and high-quality development. Using multiple panel regression models for the western region, Sun and Yuan [28] demonstrate that economic resilience contributes significantly to high-quality economic development in the short term, whereas the two have a causal relationship in the long term. Guan and Zhang [29] employ panel regression models to demonstrate how economic resilience might contribute to the Yangtze River Delta’s high-quality economic development. Most of the aforementioned research examines the impact of economic resilience in supporting high-quality development by employing panel regression models and focusing on the examination of the coupling coordination process and the current condition. Lack of driving effect for coupling coordination mechanism employing techniques such as geographical detector to measure indicator factors for each subsystem of economic resilience and high-quality economic growth. Consequently, this study presents the following three significant contributions: (1) Using the UAMRYR as a research object improves the study of urban agglomeration in central China. (2) To quantify the extent of coupling and coordination between economic resilience and high-quality economic development in the UAMRYR. (3) Combining the CCDM model and GD to precisely measure the impact values of economic resilience and economic quality development index variables to statistically examine their coupling coordination mechanism. For details, see lines 641-670 in the revised manuscript.

Comments: 6. The introduction needs to be expanded further to cover all the critical aspects of the topic. I suggest authors to consider the following articles for citations, it will strengthen the quality of this draft:

doi.org/10.3390/su14074212

doi:10.3390/ijerph16173076

doi.org/10.3390/ijerph18115697

doi.org/10.3390/ijerph18179105

doi.org/10.3390/ijerph19020958

Reply: We sincerely appreciate the valuable comments. We have carefully rewritten the introduction in the revised manuscript. Firstly, we have described the background of the study. Secondly, we have extended the literature review. And then we have summarized the previous studies. Finally, we have revealed the contribution of this study. For more details, please see lines 67-185. 

We have added the mentioned literatures in the revised manuscript. (Ref. 21, lines 878-880; Ref. 22, lines 881-883; Ref.23, lines 884-886; Ref.30, lines 908-910; Ref.31, lines 911-913)

---

## [Decision Letter · Decision Letter 1]

30 Jan 2023

Analysis of the spatio-temporal coupling coordination mechanism supporting economic resilience and high-quality 

economic development in the urban agglomeration in the middle reaches of the Yangtze River

PONE-D-22-22780R1

Dear Dr. HUANG,

We’re pleased to inform you that your manuscript has been judged scientifically suitable for publication and will be formally accepted for publication once it meets all outstanding technical requirements.

Kind regards,

Xingwei Li, Ph.D.

Academic Editor

PLOS ONE

Additional Editor Comments (optional):

Accept

Reviewers' comments:

Reviewer's Responses to Questions

**Comments to the Author**

1. If the authors have adequately addressed your comments raised in a previous round of review and you feel that this manuscript is now acceptable for publication, you may indicate that here to bypass the “Comments to the Author” section, enter your conflict of interest statement in the “Confidential to Editor” section, and submit your "Accept" recommendation.

Reviewer #1: All comments have been addressed

2. Is the manuscript technically sound, and do the data support the conclusions?

Reviewer #1: Yes

3. Has the statistical analysis been performed appropriately and rigorously? 

Reviewer #1: Yes

4. Have the authors made all data underlying the findings in their manuscript fully available?

Reviewer #1: Yes

5. Is the manuscript presented in an intelligible fashion and written in standard English?

Reviewer #1: Yes

6. Review Comments to the Author

Reviewer #1: AcceptAcceptAcceptAcceptAcceptAcceptAcceptAcceptAcceptAcceptAcceptAcceptAcceptAcceptAcceptAcceptAcceptAcceptAccept

7. PLOS authors have the option to publish the peer review history of their article (what does this mean?). If published, this will include your full peer review and any attached files.

Reviewer #1: No

---

## [Editor Report · Acceptance letter]

15 Feb 2023

PONE-D-22-22780R1 

Analysis of the spatio-temporal coupling coordination mechanism supporting economic resilience and high-quality economic development in the urban agglomeration in the middle reaches of the Yangtze River 

Dear Dr. Huang:

I'm pleased to inform you that your manuscript has been deemed suitable for publication in PLOS ONE. Congratulations! Your manuscript is now with our production department. 

Kind regards, 

on behalf of

Prof. Dr. Xingwei Li 

Academic Editor

PLOS ONE